# Inflation Method for Ensemble Kalman Filter in Soil Hydrology

Hannes H. Bauser[1,2], Daniel Berg[1,2], Ole Klein[3], and Kurt Roth[1,3]

[1]Institute of Environmental Physics (IUP), Heidelberg University, Heidelberg
[2]HGS MathComp, Heidelberg University, Heidelberg
[3]Interdisciplinary Center for Scientific Computing (IWR), Heidelberg University, Heidelberg

*Correspondence to:* Hannes H. Bauser (hannes.bauser@iup.uni-heidelberg.de)

**Abstract.** The ensemble Kalman filter (EnKF) is a popular data assimilation method in soil hydrology. In this context, it is used to estimate states and parameters simultaneously. Due to unrepresented model errors and a limited ensemble size, state and parameter uncertainties can become too small during assimilation. Inflation methods are capable of increasing state uncertainties, but typically struggle with soil hydrologic applications. We propose a multiplicative inflation method specifically designed for the needs in soil hydrology. It employs a Kalman filter within the EnKF to estimate inflation factors based on the difference between measurements and mean forecast state within the EnKF. We demonstrate its capabilities on a small soil hydrologic test case. The method is capable of adjusting inflation factors to spatiotemporally varying model errors. It successfully transfers the inflation to parameters in the augmented state, which leads to an improved estimation.

## 1    Introduction

Data assimilation combines information from models and measurements into an optimal estimate of a geophysical field of interest (Reichle, 2008). It has applications in all branches of the geosciences, with weather forecasting as the driving force behind many recent advances (van Leeuwen et al., 2015). The advantage of data assimilation methods (in contrast to e.g. inverse modeling) is the possibility to consider model errors, which are characteristic for geophysical systems.

The ensemble Kalman filter (EnKF) (Evensen, 1994; Burgers et al., 1998) is a popular data assimilation method due to its simple conceptional formulation and ease of implementation (Evensen, 2003). It is an extension of the Kalman filter (Kalman, 1960) for nonlinear models.

In hydrology, the EnKF was used for soil moisture estimation from satellite data (e.g. Reichle et al., 2002; Crow and Van Loon, 2006) or from local measurements (e.g. De Lannoy et al., 2007, 2009; Camporese et al., 2009). However, the largest uncertainties in hydrology are associated with soil hydraulic material properties. They can neither be measured directly, nor can they be transferred from the lab to the field, and are typically parameterized. Thus, including material properties into the estimation can be crucial in hydrology. Liu and Gupta (2007) called for an integrated assimilation framework including not only states but parameters, and even model structure.

The joint estimation of states and parameters in data assimilation might be one possibility to reduce the influence of model errors on parameter estimation (Liu et al., 2012). Such a joint estimation in the EnKF with an augmented state was already demonstrated by Anderson (2001) for an atmospheric model. In hydrology Vrugt et al. (2005) combined an EnKF and the

shuffled complex evolution Metropolis algorithm, while Moradkhani et al. (2005) used a dual EnKF approach to estimate states and parameters for a rainfall-runoff model. The joint assimilation of states and parameters in an augmented state was successfully performed for example in groundwater research (e.g. Chen and Zhang, 2006; Hendricks Franssen and Kinzelbach, 2008; Kurtz et al., 2012, 2014; Erdal and Cirpka, 2016), but also in soil hydrology for land surface models (e.g. Bateni and
Entekhabi, 2012; Han et al., 2014; Zhang et al., 2017) and on smaller scales based on the Richards equation (e.g. Li and Ren, 2011; Wu and Margulis, 2011, 2013; Song et al., 2014; Erdal et al., 2014, 2015; Shi et al., 2015; Bauser et al., 2016; Brandhorst et al., 2017; Botto et al., 2018).

Due to unrepresented model errors and due to a limited ensemble size, the EnKF underestimates model errors, which can lead to filter inbreeding. Systematic model errors are common for example in land surface models (Vereecken et al., 2015).
Additionally, in soil hydrology spatially and temporally varying model errors occur due to un- or ill-represented processes like preferential flow or hysteresis. Underestimated errors cause an insufficient ensemble spread in the augmented state. This is especially severe for parameters, which are typically not changed through a forward propagation and consequently cannot increase their uncertainty again. Due to the convergent dynamics in soil hydrology, the uncertainty in the state depends strongly on the parameter spread and becomes too small as well.

Covariance inflation can counteract filter inbreeding. Different methods have been proposed: (i) Additive inflation, which adds a model error after the forward propagation. This method is especially useful if prior knowledge about the model error exists. In atmospheric sciences additive inflation has been successfully applied by e.g. using reanalysis of historical weather prediction errors (Whitaker et al., 2008). (ii) Relaxation methods, which relax the analysis back to a prior perturbation or spread, have been proposed with tuning parameters (Zhang et al., 2004; Whitaker and Hamill, 2012) or based on deviations to
measurements (Ying and Zhang, 2015). (iii) Multiplicative covariance inflation, which inflates the complete state with a scalar factor, where the inflation factor is either chosen manually (Anderson and Anderson, 1999) or is estimated based on deviations from measurements (e.g. Wang and Bishop, 2003; Anderson, 2007; Li et al., 2009). This method has been further extended to inflate each state component individually (Anderson, 2009).

All these inflation methods are developed in an atmospheric sciences context. Their transfer to soil hydrology is limited,
due to the spatiotemporally varying model errors and the typically employed augmented state. For groundwater research, Kurtz et al. (2012) reported improved results by employing the inflation method by Anderson (2007), and Kurtz et al. (2014) used the constant inflation by Anderson and Anderson (1999). In soil hydrology, however, adjusted methods have been used: For example Han et al. (2014) and Zhang et al. (2017) apply a special case of the inflation method by Whitaker and Hamill (2012) and keep the parameter spread constant to ensure a sufficient ensemble spread. Bauser et al. (2016) used the method by
Anderson (2009), but adjusted the inflation of parameters.

Alternatively, no inflation method is reported (e.g. Li and Ren, 2011; Shi et al., 2015), but instead a damping factor (Hendricks Franssen and Kinzelbach, 2008), which can alleviate the issue, is employed. This is done by e.g. Wu and Margulis (2011); Song et al. (2014); Erdal et al. (2014); Brandhorst et al. (2017); Botto et al. (2018), where Erdal et al. (2014) and Brandhorst et al. (2017) combined this method with additive inflation.

In this paper, we propose a novel multiplicative inflation method, specifically designed for the needs in the soil hydrology community. The inflation method can vary rapidly in space and time to cope with the typically varying model errors and it is capable of a transfer of the inflation in the state to the parameters in the augmented state. The remainder of this paper is organized as follows: Sect. 2 describes (i) the EnKF, (ii) our proposed inflation method and (iii) a soil hydrologic test case.

Section 3 shows the results of our method applied to the test case, followed by discussion and conclusion in Sect. 4 and 5.

## 2 Method

### 2.1 Ensemble Kalman Filter

The EnKF (Evensen, 1994; Burgers et al., 1998) is the Monte Carlo extension of the Kalman filter (Kalman, 1960) for nonlinear models and assumes unbiased Gaussian error distributions to combine model and measurement information. The filter is a

sequential method and alternates between a forecast step and an analysis step. The forecast propagates a state including its uncertainty forward in time. The analysis combines uncertain model information with uncertain measurements at this time into an optimal estimate of the state. These two steps are now explained in more detail.

The forecast propagates an ensemble of states $\varphi^n$ forward from time $k-1$ to time $k$ with a model $M$,

$$\varphi_k^{\mathrm{f},n} = M(\varphi_{k-1}^{\mathrm{a},n}) + \beta^n \,, \tag{1}$$

where the superscripts f and a denote forecast and analysis respectively, while $n$ denotes the ensemble members with $n = 1, ..., N$. The uncertainty in the state is directly represented through the ensemble $\varphi_{k-1}^{\mathrm{a},n}$ and then propagated nonlinearly with the model. Unrepresented model errors can be added through the unbiased Gaussian process noise $\beta$. This is also called additive inflation. However, the details of the model error are typically unknown and thus not represented adequately. The propagated uncertainties are directly represented through the new forecast ensemble $\varphi_k^{\mathrm{f},n}$.

The state can be extended by e.g. model parameters $\phi$ to an augmented state $\boldsymbol{u} = [\varphi, \phi]$. This requires a forecast for each augmented state component. Parameters are typically assumed to be constant in time:

$$\phi_k^{\mathrm{f},n} = \phi_{k-1}^{\mathrm{a},n} \,. \tag{2}$$

The forecast of the state $\varphi_k^{\mathrm{f},n}$ now also depends on the corresponding parameter set $\phi_{k-1}^{\mathrm{a},n}$. This way, uncertainties in the parameters are propagated as well and can be reduced jointly in the analysis.

Assuming unbiased Gaussian distributions, the ensemble of augmented states is characterized through the forecast error covariance matrix $\mathbf{P}^{\mathrm{f}}$,

$$\mathbf{P}_k^{\mathrm{f}} = \overline{\left[ \boldsymbol{u}_k^{\mathrm{f},n} - \overline{\boldsymbol{u}_k^{\mathrm{f}}} \right] \left[ \boldsymbol{u}_k^{\mathrm{f},n} - \overline{\boldsymbol{u}_k^{\mathrm{f}}} \right]^T} \,, \tag{3}$$

where the overline denotes the expectation value and $\overline{\boldsymbol{u}_k^{\mathrm{f}}}$ is the ensemble mean.

The analysis combines model and measurement information based on the Gaussian error assumption. The measurement error covariance matrix $\mathbf{R}$ of the measurements $d$ is defined analogously as

$$\mathbf{R}_k = \overline{[\boldsymbol{\epsilon}_k^n][\boldsymbol{\epsilon}_k^n]^T}, \tag{4}$$

where $\boldsymbol{\epsilon}$ is the measurement error. The measurements are linked to the state through the linear measurement operator $\mathbf{H}$, which maps from the state space to the measurement space:

$$d_k = \mathbf{H}_k u_{\text{true},k} + \boldsymbol{\epsilon}_k. \tag{5}$$

The Kalman gain $\mathbf{K}$ weighs the forecast error covariance matrix with the measurement error covariance matrix and maps from the measurement space back to the state space, based on the covariances in the forecast error covariance matrix:

$$\mathbf{K}_k = \mathbf{P}_k^{\text{f}} \mathbf{H}_k^T \left[\mathbf{H}_k \mathbf{P}_k^{\text{f}} \mathbf{H}_k^T + \mathbf{R}_k\right]^{-1}. \tag{6}$$

Based on the measurements, the Kalman gain updates the forecast ensemble to the analysis ensemble:

$$\boldsymbol{u}_k^{\text{a},n} = \boldsymbol{u}_k^{\text{f},n} + \mathbf{K}_k \left[d_k + \boldsymbol{\epsilon}_k^n - \mathbf{H}_k \boldsymbol{u}_k^{\text{f},n}\right]. \tag{7}$$

This update to the ensemble $\boldsymbol{u}_k^{\text{a},n}$ minimizes the analysis error covariance $\mathbf{P}_k^a$, which fulfills

$$\mathbf{P}_k^{\text{a}} = [\mathbf{I} - \mathbf{K}_k \mathbf{H}_k]\,\mathbf{P}_k^{\text{f}}, \tag{8}$$

for infinite ensemble sizes.

Through spurious correlations and non-Gaussian distributions, $\mathbf{P}_k^{\text{a}}$ will become too small, which can lead to filter inbreeding and ultimately filter divergence (e.g. Hamill et al., 2001). This is intensified, if the model error required in Eq. (1) is unknown. A common way to alleviate this issue in hydrology is the use of a damping factor $\gamma \in [0,1]$ (Hendricks Franssen and Kinzelbach, 2008), which is multiplied to the correction vector in Eq. (7) and consequently lessens the uncertainty reduction. The damping factor can be extended to a vector $\boldsymbol{\gamma}$ (and an entrywise multiplication) to treat augmented state components differently (Wu and Margulis, 2011). Typically, parameters are multiplied with a smaller factor than the state. However, the damping factor can only alleviate and not completely prevent the inbreeding problem.

## 2.2 Multiplicative inflation for soil hydrology

Multiplicative inflation is another heuristic way to avoid filter inbreeding. Anderson and Anderson (1999) proposed to increase the distance of each ensemble member to the ensemble mean by multiplying this distance by $\sqrt{\lambda}$ for the inflation factor $\lambda \geq 1$:

$$\boldsymbol{u}_{\text{inf}}^{\text{f},n} = \sqrt{\lambda}\left(\boldsymbol{u}^{\text{f},n} - \overline{\boldsymbol{u}^{\text{f}}}\right) + \overline{\boldsymbol{u}^{\text{f}}}. \tag{9}$$

This inflation factor is applied to the complete augmented state and has to be adjusted to the specific problem. By construction, it does not alter the mean value: $\overline{\boldsymbol{u}_{\text{inf}}^{\text{f}}} = \overline{\boldsymbol{u}^{\text{f}}}$. A temporally varying inflation factor can be estimated by comparing uncertainties with

the distance of measurement and forecast (e.g. Wang and Bishop, 2003; Anderson, 2007; Li et al., 2009). A spatiotemporally adaptive inflation has been achieved by estimating a vector $\boldsymbol{\lambda}$ for the complete augmented state (Anderson, 2009). The author uses the correlation between measurements and augmented state dimensions and asks the question: *How much inflation is required in each dimension to explain the observed differences to the measurements?* Anderson (2009) showed that this works excellently for the actual state. However, we experienced possible over-inflation in parameters (which do not have any dynamics to compensate for this), which can lead to filter collapses.

We propose a more conservative inflation method and ask the question: *How much of the required change of the inflation are we allowed to transfer to the state dimensions based on the correlation information?* This can be achieved by applying a Kalman filter for the inflation within the EnKF.

In this Kalman filter, the inflation vector is treated as the state variable. As for parameters, we choose a constant model for the forecast in time:

$$\boldsymbol{\lambda}_k^{\mathrm{f}} = \boldsymbol{\lambda}_{k-1}^{\mathrm{a}}. \tag{10}$$

For convenience we will drop the time subscript $k$ in the following. Furthermore, we will use the same symbols as for the EnKF, but denote them with the subscript $\lambda$. We approximate the forecast error covariance matrix for lambda, $\mathbf{P}_\lambda^{\mathrm{f}}$, based on the covariance matrix of the augmented state in the EnKF, $\mathbf{P}^{\mathrm{f}}$, as the normalized absolute correlation matrix of the augmented state ensemble. The matrix component $ij$ is determined as

$$\left(\mathbf{P}_\lambda^{\mathrm{f}}\right)_{ij} = \sigma_\lambda^2 \left|\left(\mathbf{P}^{\mathrm{f}}\right)_{ij}\right| \left[\left(\mathbf{P}^{\mathrm{f}}\right)_{ii}\left(\mathbf{P}^{\mathrm{f}}\right)_{jj}\right]^{-\frac{1}{2}}, \tag{11}$$

where $\sigma_\lambda$ denotes the uncertainty of the inflation factors. It is a tuning parameter that is kept constant over time and is assigned to all state dimensions. It influences how fast the inflation factors are adjusted. This follows the idea by Anderson (2007, 2009) to avoid a closure problem, where the inflation estimation would require its own inflation. Instead, the uncertainty is kept constant. Furthermore, only the absolute value of the correlation is considered, since the inflation is based on differences between measurement and model, but ignores their direction. Note, that this presumes that the correlations of the model state can be transferred to the inflation. In the presence of unknown model errors this assumption may or may not be correct. However, the estimation at measurement locations will remain meaningful in any case.

For the analysis, the distance $\boldsymbol{d}_\lambda$ between mean forecast and measurement is used as measurement for $\boldsymbol{\lambda}$:

$$\boldsymbol{d}_\lambda = \left|\boldsymbol{d} - \mathbf{H}\overline{\boldsymbol{u}_{\mathrm{inf}}^{\mathrm{f}}}\right|. \tag{12}$$

The measurement error covariance matrix $\mathbf{R}_\lambda$ of $\boldsymbol{d}_\lambda$ can be calculated based on the error covariance matrices of $\boldsymbol{d}$ and $\mathbf{H}\overline{\boldsymbol{u}_{\mathrm{inf}}^{\mathrm{f}}}$,

$$\left(\mathbf{R}_\lambda\right)_{ij} = \left|\left(\mathbf{R}\right)_{ij} + \left(\mathbf{H}\mathbf{P}_{\mathrm{inf}}^{\mathrm{f}}\mathbf{H}^T\right)_{ij}\right|, \tag{13}$$

where the inflated forecast error covariance matrix $\mathbf{P}_{\mathrm{inf}}^{\mathrm{f}}$ can be calculated from the inflation vector and the forecast error covariance matrix by combining Eq. (9) (with vector lambda and entrywise multiplication) and Eq. (3): $\mathbf{P}_{\mathrm{inf}}^{\mathrm{f}} = \mathbf{P}^{\mathrm{f}} \circ [\sqrt{\boldsymbol{\lambda}^{\mathrm{f}}}\sqrt{\boldsymbol{\lambda}^{\mathrm{f}}}^T]$. The entrywise product is denoted by $\circ$ and the entrywise square root of $\boldsymbol{\lambda}$ by $\sqrt{\boldsymbol{\lambda}}$.

The expected distance between measurement and mean forecast based on the current inflation is

$$\left( \boldsymbol{h}_\lambda(\boldsymbol{\lambda}^{\mathrm{f}}) \right)_i = [(\mathbf{R}_\lambda)_{ii}]^{\frac{1}{2}}, \tag{14}$$

which combines the uncertainties of $\boldsymbol{d}$ and $\overline{\mathbf{H} \boldsymbol{u}_{\mathrm{inf}}^{\mathrm{f}}}$. To be able to determine the Kalman gain, we first calculate the Jacobian matrix $\mathbf{H}_\lambda$ of partial derivatives of $\boldsymbol{h}_\lambda$ with respect to $\boldsymbol{\lambda}$:

$$(\mathbf{H}_\lambda)_{ij} = \frac{\partial}{\partial \left( \boldsymbol{\lambda}^{\mathrm{f}} \right)_j} \left( \boldsymbol{h}_\lambda(\boldsymbol{\lambda}^{\mathrm{f}}) \right)_i, \tag{15}$$

$$= \left[ 2 \left[ (\boldsymbol{\lambda}^{\mathrm{f}})_j \right]^{\frac{1}{2}} \left( \boldsymbol{h}_\lambda(\boldsymbol{\lambda}^{\mathrm{f}}) \right)_i \right]^{-1} \sum_m (\mathbf{H})_{ij} (\mathbf{H})_{im} \left( \mathbf{P}^{\mathrm{f}} \right)_{jm} \left[ \left( \boldsymbol{\lambda}^{\mathrm{f}} \right)_m \right]^{\frac{1}{2}}. \tag{16}$$

With this approximated measurement operator $\mathbf{H}_\lambda$, the Kalman gain $\mathbf{K}_\lambda$ and the analysis state $\boldsymbol{\lambda}^{\mathrm{a}}$ are obtained as

$$\mathbf{K}_\lambda = \mathbf{P}_\lambda^{\mathrm{f}} \mathbf{H}_\lambda^T \left[ \mathbf{H}_\lambda \mathbf{P}_\lambda^{\mathrm{f}} \mathbf{H}_\lambda^T + \mathbf{R}_\lambda \right]^{-1}, \tag{17}$$

$$\boldsymbol{\lambda}^{\mathrm{a}} = \boldsymbol{\lambda}^{\mathrm{f}} + \mathbf{K}_\lambda \left[ \boldsymbol{d}_\lambda - \boldsymbol{h}_\lambda(\boldsymbol{\lambda}^{\mathrm{f}}) \right]. \tag{18}$$

Note, that the matrices $\mathbf{P}_\lambda^{\mathrm{f}}$ and $\mathbf{R}_\lambda$ can possibly become indefinite, due to the absolute value in Eq. (11) and Eq. (13). Consequently, the inverse in Eq. (17) could become unfeasible. However, we never encountered such a case. In a situation with uncorrelated measurements, the issue can be resolved by reducing $\sigma_\lambda$ just for that single time step.

With this Kalman filter, the inflation vector is updated at each time step based on the difference of the mean forecast to the measurements. Following Anderson (2007), we additionally prohibit a deflation by constraining the inflation values to
$(\boldsymbol{\lambda})_i \geq 1$.

## 2.3  Model

We test the proposed inflation method on a small hydrologic test case. We constructed it specifically to require a strong inflation. This makes it possible to explore features of the inflation in detail on a rather short timescale. Due to a small ensemble size, the results vary depending on the seed of the random numbers. This however, is related to different performance of the EnKF
itself. In simulations (results are not shown), we found that the behavior of the inflation remains consistent. We have also tested the inflation method with real-world data by reanalyzing the application by Bauser et al. (2016) with the main result shown in Appendix B.

The Richards equation describes the change of volumetric soil water content $\theta$ (–) in a continuous porous medium,

$$\frac{\partial \theta}{\partial t} - \nabla \cdot \left[ K(\theta) \left[ \nabla h_{\mathrm{m}}(\theta) - 1 \right] \right] = 0, \tag{19}$$

where $K$ (L T$^{-1}$) is the isotropic conductivity and $h_{\mathrm{m}}$ (L) is the matric head. Both are related to the water content. This relation is typically described through parameterized material properties. We choose the Mualem–van Genuchten parameterization

(Mualem, 1976; van Genuchten, 1980),

$$K(\Theta) = K_0 \Theta^\tau \left[ 1 - \left[ 1 - \Theta^{n/[n-1]} \right]^{1-1/n} \right]^2 , \tag{20}$$

$$h_m(\Theta) = \frac{1}{\alpha} \left[ \Theta^{-n/[n-1]} - 1 \right]^{1/n} , \tag{21}$$

with the saturation $\Theta$ (–),

$$\Theta := \frac{\theta - \theta_{\mathrm{r}}}{\theta_{\mathrm{s}} - \theta_{\mathrm{r}}} . \tag{22}$$

The parameterization is described by a set of six parameters: $\theta_{\mathrm{s}}$ (–), $\theta_{\mathrm{r}}$ (–), $\alpha$ ($\mathrm{L}^{-1}$), $n$ (–), $K_0$ ($\mathrm{L\,T}^{-1}$) and $\tau$ (–).

We additionally consider small scale heterogeneity through Miller scaling. It assumes geometrical similarity. With this the microscopic geometry of the pore space at a macroscopic position is parameterized by a single length scale $\xi$ and the macroscopic heterogeneity field can be generated with a single scalar field of this length scale. Miller and Miller (1956) showed that the hydraulic functions scale with this parameter according to

$$K(\theta) = K^*(\theta) \cdot \xi^2 , \tag{23}$$

$$h_{\mathrm{m}}(\theta) = h_{\mathrm{m}}^*(\theta) \cdot \frac{1}{\xi} , \tag{24}$$

where the functions $K(\theta)$ and $h_m(\theta)$ are defined at a reference point $^*$ with Miller scaling parameter $\xi = 1$ and from there are projected to all locations.

For the test, we choose a one-dimensional case with a depth of $50\,\mathrm{cm}$ for a time of $6\,\mathrm{days}$. We set a groundwater table as the lower boundary condition throughout the whole time and start from equilibrium conditions. The upper boundary condition is no flux, except for a rain event with $2.0 \cdot 10^{-7}$ $[\mathrm{m\,s}^{-1}]$ during the fourth day. As observations we choose two water content measurements at a depth of $9.5\,\mathrm{cm}$ and $19.5\,\mathrm{cm}$ as they would be available from time domain reflectometry (TDR). We set the measurement uncertainty to a standard deviation of $0.007$ (e.g. Jaumann and Roth, 2017).

As material we choose sandy loam from Carsel and Parrish (1988): $\theta_{\mathrm{s}} = 0.41$, $\theta_{\mathrm{r}} = 0.065$, $\alpha = -7.5\,\mathrm{m}^{-1}$, $n = 1.89$, $K_0 = 1.23 \cdot 10^{-5}\,\mathrm{m\,s}^{-1}$ and $\tau = 0.5$. For the Miller scaling we choose $\xi_1 = 0.32$ at the upper measurement position and $\xi_2 = 3.2$ at the lower measurement position. We reduce the description of the heterogeneity to these two parameters. The full function of the scaling factors is calculated by linearly interpolating between the measurement positions and constantly extrapolating to the boundaries.

The forward simulations are performed using MuPhi (Ippisch et al., 2006) with a spatial resolution of $1\,\mathrm{cm}$. This corresponds to a state with 50 dimensions.

To test the inflation method, we perform a perfect model experiment. With the EnKF we estimate the water content state and four parameters ($\xi_1$, $\xi_2$, $K_0$ and $\tau$) through the augmented state $\boldsymbol{u} = [\boldsymbol{\theta}, \log_{10}(\xi_1), \log_{10}(\xi_2), \log_{10}(K_0), \tau]$. We choose to include the logarithm of $\xi_1$, $\xi_2$ and $K_0$, because we expect a more linear relation to the water content state, than for the actual parameters. For the water content state, we use the correct initial condition as the mean with an uncertainty of $0.005$. The uncertainty is spatially correlated using the fifth-order piecewise rational function by Gaspari and Cohn (1999) with the length-scale $c = 5\,\mathrm{cm}$. As initial guess for the parameters, we start with unknown heterogeneity $\log_{10}(\xi_1) = \log_{10}(\xi_2) = 0.0 \pm 0.25$,

corresponding to two standard deviations away from the true values of $\log_{10}(\xi_1) = -0.5$ and $\log_{10}(\xi_2) = 0.5$. For the saturated hydraulic conductivity, we choose a too small value of $\log_{10}(K_0) = -5.5 \pm 0.5$, $K_0$ in $(\mathrm{m\,s^{-1}})$, which is about one standard deviation away from the true value of $\log_{10}(K_0) = -4.9$. For the tortuosity $\tau = 0.5 \pm 0.5$ we start from the true value.

Through the unrepresented heterogeneity, we can mimic a model error, leading to a bias towards smaller values for the estimation of $K_0$ during times without dynamics, which may necessitate inflation. The parameter $\tau$ is expected to have a smaller influence, since the uncertainty is chosen small and it is already at the true value. This way it can act as an indicator parameter for the inflation as it does not require inflation.

The EnKF is set up with a total of 25 ensemble members and a damping vector of $\gamma = [\mathbf{1.0}, 0.3, 0.3, 0.3, 0.3]$, which we also apply to the inflation. The damping factor of 0.3 is applied to the parameters to alleviate issues of nonlinear relations between observations and parameters. For the uncertainty of the inflation factors we choose $\sigma_\lambda = 1.0$.

## 3 Results

We estimate the water content state together with the four parameters $\xi_1$, $\xi_2$, $K_0$ and $\tau$ with the EnKF as described in Sect. 2.3. The development of the water content at the two measurement locations at a depth of 9.5 cm and 19.5 cm is shown together with the inflation factor at these locations in Fig. 1. The inflation factor is applied to the forecast ensemble before the analysis. The standard deviation of the inflated ensemble should describe the distance of the estimated mean to the synthetic truth. Note, that the inflation factor is not based on this distance and relies on the noisy measurements. Therefore, it is only an indicator.

During the first three days without any dynamics, the uncertainty for the upper measurement is slightly underestimated, while the uncertainty in the lower measurement is slightly overestimated. This leads to an inflation factor of basically 1 for the lower measurement (factors smaller than 1 are not allowed), while the inflation factor for the upper measurement is larger. However, due to correlations between the measurement locations a stronger inflation to fully explain the difference to the truth is prevented.

The deviation from the synthetic truth is induced through the initial guess of no heterogeneity and can also be seen in the systematic deviation of the inflated mean (which is equal to the forecast mean) from the analysis mean. When the infiltration front reaches the measurements, the deviations from the truth, underestimation of the uncertainty, and inflation factors increase rapidly. All of them are more pronounced for the upper measurement location. After the main peak, the differences and also the inflation factors decrease rapidly again.

The inflation factor for the state is shown in Fig. 2. It shows the strong increase of the inflation factor during the infiltration and its fast decrease afterwards. The inflation is strongest at the measurement location at a depth of 9.5 cm. The inflation factor is transferred to the other state locations through the correlations, which decrease with distance. Directly below the measurement locations the inflation factors are increased less than above. This is due to the chosen interpolation of the Miller scaling factors. Through the interpolation between the measurement locations and extrapolation to the boundaries changes, the dynamics changes at the measurement locations. During the infiltration the dynamics is mainly influenced by the water content above and the correlations to these locations are stronger.

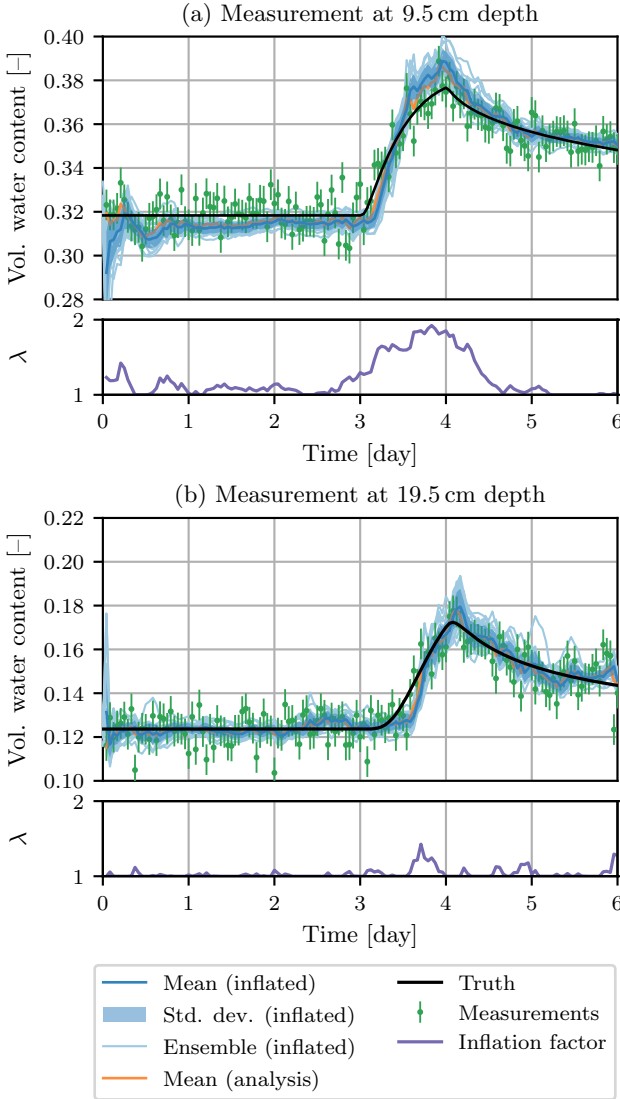

**Figure 1.** Water content estimation at the two measurement locations. The standard deviation of the inflated ensemble should be able to explain the differences between the inflated mean and the synthetic truth. The inflation factor is increased, when the ensemble uncertainty is too small.

The development of the Miller scaling factors $\xi_1$ and $\xi_2$ at the two measurement positions (9.5 cm and 19.5 cm depth) is shown in Fig. 3(a) and 3(b) together with the estimated inflation factor for these parameters. Both initial conditions assume no heterogeneity and start at $\log_{10}(\xi_1) = \log_{10}(\xi_2) = 0.0 \pm 0.25$, corresponding to two standard deviations away from the true value. At the upper location the true value of $\log_{10}(\xi_1) = -0.5$ corresponds to a finer material. Consequently, the water content drops, as seen in Fig. 1, leading to a strong correlation with the scaling factor, and $\log_{10}(\xi_1)$ is adjusted rapidly to lower

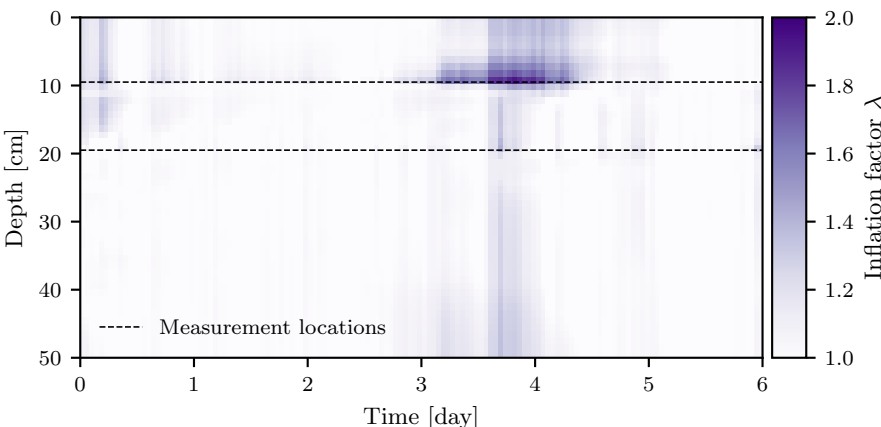

**Figure 2.** Inflation factor for the water content state. The inflation is strongest at the upper measurement location during the infiltration, when the uncertainty is underestimated the most. The inflation factor is transferred to the other measurement locations through the correlations in the Kalman gain. The used interpolation of Miller scaling factors impacts these correlations and leads to the smaller inflation directly below the measurement locations.

values. Accordingly, the inflation factor is increased quickly in the beginning and then reduced back to 1 when the estimation of $\log_{10}(\xi_1)$ reaches and eventually underestimates the true value. The underestimation of the scaling factor corresponds to a too fine material, which leads to slower changes in the water content state and therefore smaller correlations. The scaling factor is corrected during the rain event on the fourth day, which also leads to an inflation.

5    The initial guess for the scaling factor for the depth of $19.5\,\mathrm{cm}$ underestimates the scaling factor, which corresponds to a too fine material. Again, the correlations are small. The value increases slowly during the dry period in the beginning, but is inflated and adjusted strongly during the rain event.

    The saturated hydraulic conductivity $K_0$ (Fig. 3(c)) was chosen to start a little more than one standard deviation below the true value. Due to the unrepresented heterogeneity in the beginning, the value decreases even further. The inflation remains 10    very small due to correlations to both measurement locations. However, as soon as the infiltration event reaches the first measurement location, the value is corrected towards the true value. At the same time, the inflation factor is increased due to the too small uncertainty. After the rain event the inflation factor drops rapidly back to one. The hydraulic conductivity remains below the true value. Another rain event would be required to improve the estimation further.

    The tortuosity $\tau$ (Fig. 3(d)) also influences the hydraulic conductivity function, but has in this case much smaller impact and 15    consequently smaller correlations to the measurements than $K_0$. We use it as an indicator parameter and start at the true value. During the infiltration event the value is changed due to its correlation. The corresponding inflation factor is increased as well, but remains small enough and drops back to 1 quickly enough to not cause any over-inflation of the parameter.

    To emphasize the need of a fast adapting inflation factor, we reduce the uncertainty of the inflation factors to $\sigma_\lambda = 0.5$ to slow down their adjustment. The results are summarized in Fig. 4. The inflation of the water content state (Fig. 4(a)) shows,

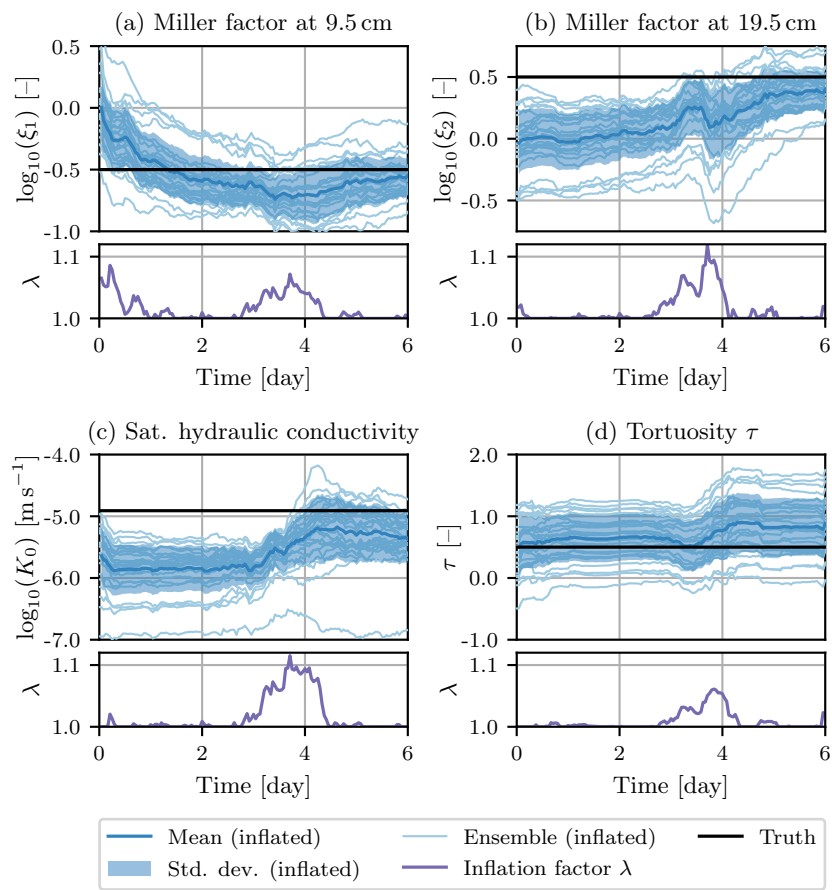

**Figure 3.** Development of Miller scaling factors $\xi_1$ and $\xi_2$, saturated hydraulic conductivity $K_0$ and tortuosity $\tau$ together with their corresponding inflation factors during estimation with the EnKF.

that the inflation factor does not reach as high values as before (see Fig. 2). To compensate for this, the inflation acts over a longer period of time. The same effect is also observed in the inflation of the parameters (Fig. 4(b) and (c)). This leads to a smaller inflation during the rain event and consequently a too small uncertainty. At later times, when the cause of the error is not active any more, the correlations to measurement locations are reduced leading to a slower reduction of the inflation in the

5 parameters. In the indicator parameter $\tau$ the beginning of an over-inflation can be seen towards later times. This necessitates a more rapid inflation when correlations are used to update inflation information.

The results for the parameters $K_0$ and $\tau$ of a run without inflation (and only damping) are shown in Fig. 5. Again, $K_0$ moves further away from the true value due to the unrepresented heterogeneity and comes closer to the true value during the infiltration event. However, since the Miller scaling factor is not inflated in the beginning, it is adjusted slower. Consequently,

10 the $K_0$ is corrected longer in the wrong direction. The uncertainty eventually becomes too small and in the end the mean is more than 5 standard deviations away from the true value, since the uncertainty cannot be increased any more.

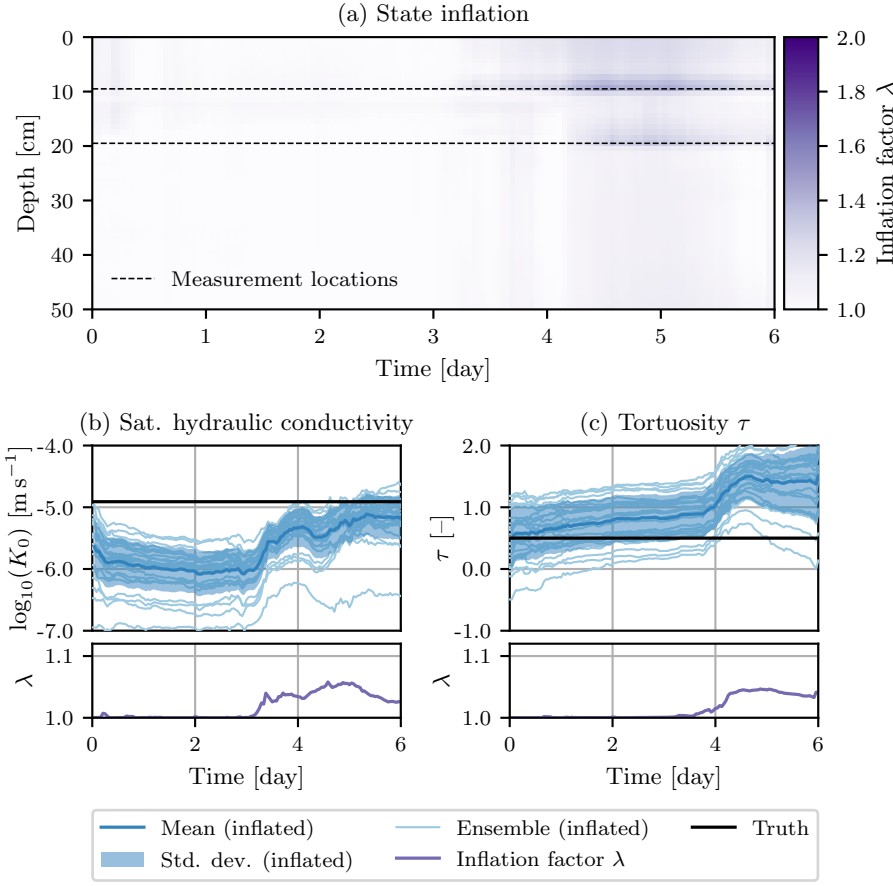

**Figure 4.** Development of the inflation factor for the water content state and saturated hydraulic conductivity $K_0$ and tortuosity $\tau$ together with their corresponding inflation factors for an estimation with a reduced inflation factor uncertainty of $\sigma_\lambda = 0.5$.

## 4 Discussion

The proposed inflation method uses a Kalman filter to estimate inflation factors within the EnKF. It is based on the difference between measurements and mean forecast state. It transfers correlations from the forecast of the augmented state to the inflation. Consequently, the performance will be limited if model errors are structurally not represented in the forecast error covariance matrix. The estimation of the inflation factors with a Kalman Filter is, like the EnKF itself, based on a linearized analysis. The use of a damping factor can alleviate issues with estimating nonlinear dependent parameters. To keep the inflation consistent with the analysis in the EnKF, we apply the same damping factor for both.

We designed a small synthetic hydrologic test case for the inflation. This test case mimics a model error through initially unrepresented heterogeneity. We designed the test case so that a strong temporally varying inflation is necessary, as it can occur with real data. We choose a short time so that the details of the behavior of the method can be explored. The method showed

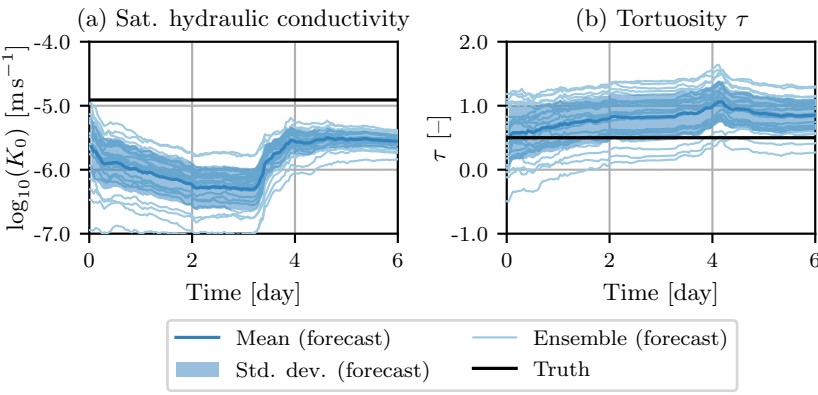

**Figure 5.** Development of saturated hydraulic conductivity $K_0$ and tortuosity $\tau$ for an estimation without inflation.

that it is capable of inflating states and parameters. The inflation is adjusted fast and differentiates between parameters with strong and not so strong correlations. No over-inflation of weakly correlated parameters occurred. In this specific test case the estimation with inflation is far superior to an estimation without inflation.

The fast adjustment speed of the inflation factor is important because of the fast changing model errors and correlations to
parameters. The adjustment speed is determined by the uncertainty of the inflation factor. This uncertainty is set to a constant value and has to be adjusted. For all our cases a value of $\sigma_\lambda = 1$ was sufficient, but larger values were possible too. The need for such a fast adjustment is shown by estimating the same case with a reduced uncertainty of $\sigma_\lambda = 0.5$, which leads to a slower adaptation of the inflation factor. This leads to smaller inflation factors, which is compensated by maintaining them for a longer period of time. In this test case this leads to inflation at times after the infiltration front has passed the measurements already
and the model error is small again. This can cause over-inflation of weakly correlated parameters. Too large uncertainties of the inflation (in our test case $\sigma_\lambda = 4$), where the uncertainty is larger than the typical range for the values of lambda, can also lead to overinflation of weakly correlated parameters. Reasons for this can be the linearizations in the analysis and the calculation of the Jacobian (Eq. 15). This limits the adjustment speed of the inflation.

Fast dropping correlations between measurements and parameters are a limit for the method. An example could be a param-
eter only acting on an infiltration boundary condition. After the infiltration is over, correlations to this parameter would drop to zero and the inflation factor for this parameter will not be changed any more. If the inflation factor is not equal to 1 at this time, the parameter spread will keep increasing. In such a case, when there is no correlation, the parameter should be excluded from the estimation and consequently also from the inflation.

The method is in principle capable of compensating unrepresented model errors. However, it relies on correlations cal-
culated from the forecast ensemble of the augmented state. If parameters have correlations to measurement locations with underestimated forecast uncertainties, the inflation will keep increasing the parameter spread until the forecast uncertainties are increased sufficiently. Therefore the correlations have to contain useful information. This means that inflating the parameters based on their correlations to measurement locations has to increase the forecast spread at these measurement locations. If

the parameters have an insufficient influence on the state uncertainty an over-inflation of the parameters can occur. An example are measurements with underestimated measurement uncertainties and short time between measurements compared to the timescale of the dynamics. Then the parameters are not able to increase the state uncertainty in the short forecast time between measurements and the forecast dynamics is not able represent the measurement noise. If such errors occur intermittently, e.g.,

the closed-eye period as proposed by Bauser et al. (2016) could be used to bridge these times. A rather heuristic solution could be a decay of the inflation factor towards values of 1, as already proposed by Anderson (2009).

## 5  Conclusions

In this work we propose a novel spatiotemporally adaptive inflation method, specifically designed for soil hydrology, which nevertheless is expected to work in similar systems as well. The inflation method is based on a Kalman filter acting within the

EnKF. The method is capable of rapid adjustments of inflation factors, treating each augmented state dimension individually. This rapid adjustment is required due to temporally varying model errors, as they can appear through violation of the local equilibrium assumption of the Richards equation, hysteresis, or unrepresented heterogeneity.

We demonstrate the use of our inflation method in combination with a damping factor on a small hydrologic example. We choose heterogeneity as a possible model error, but allow the heterogeneity to be estimated along with the soil hydrologic

parameters $K_0$ and $\tau$ of the Mualem–van Genuchten parameterization. Our proposed inflation method proved to be stable in combination with parameter estimation. The performance of the estimation improved and parameter uncertainty remained consistent. The method requires that the correlations from in the forecast ensemble contain useful information for the inflation. However, we demonstrate that it even works for only weakly correlated parameters. We expect the inflation method to generally improve data assimilation with the EnKF and to thus lead to better state and parameter estimations in soil hydrology.

**Appendix A:  Jacobian in the inflation method**

We briefly show the derivation of the Jacobian matrix $\mathbf{H}_\lambda$ for the inflation (Eq. 15). Again, the entrywise product is denoted by $\circ$ and the entrywise square root of $\boldsymbol{\lambda}$ by $\sqrt{\boldsymbol{\lambda}}$:

$$(\mathbf{H}_\lambda)_{ij} = \frac{\partial}{\partial\left(\boldsymbol{\lambda}^{\mathrm{f}}\right)_j}\left(\boldsymbol{h}_\lambda(\boldsymbol{\lambda}^{\mathrm{f}})\right)_i$$

$$= \frac{\partial}{\partial(\boldsymbol{\lambda}^{\mathrm{f}})_j}\left[(\mathbf{R})_{ii} + \left(\mathbf{H}\left[\mathbf{P}^{\mathrm{f}}\circ\left[\sqrt{\boldsymbol{\lambda}^{\mathrm{f}}}\sqrt{\boldsymbol{\lambda}^{\mathrm{f}}}^T\right]\right]\mathbf{H}^T\right)_{ii}\right]^{\frac{1}{2}}, \text{ with } \mathbf{P}^{\mathrm{f}} \text{ symmetric}$$

$$= \frac{\partial}{\partial(\boldsymbol{\lambda}^{\mathrm{f}})_j}\left[(\mathbf{R})_{ii} + \sum_m\sum_k(\mathbf{H})_{im}(\mathbf{H})_{ik}\left(\mathbf{P}^{\mathrm{f}}\right)_{km}\left[(\boldsymbol{\lambda}^{\mathrm{f}})_m\right]^{\frac{1}{2}}\left[(\boldsymbol{\lambda}^{\mathrm{f}})_k\right]^{\frac{1}{2}}\right]^{\frac{1}{2}}$$

$$= \left[2\left(\boldsymbol{h}_\lambda(\boldsymbol{\lambda}^{\mathrm{f}})\right)_i\right]^{-1}\sum_m\sum_k(\mathbf{H})_{im}(\mathbf{H})_{ik}\left(\mathbf{P}^{\mathrm{f}}\right)_{km}\frac{\partial}{\partial(\boldsymbol{\lambda}^{\mathrm{f}})_j}\left[\left[(\boldsymbol{\lambda}^{\mathrm{f}})_m\right]^{\frac{1}{2}}\left[(\boldsymbol{\lambda}^{\mathrm{f}})_k\right]^{\frac{1}{2}}\right]$$

$$= \left[2\left(\boldsymbol{h}_\lambda(\boldsymbol{\lambda}^{\mathrm{f}})\right)_i\right]^{-1}\sum_m\sum_k(\mathbf{H})_{im}(\mathbf{H})_{ik}\left(\mathbf{P}^{\mathrm{f}}\right)_{km}\frac{1}{2}\left[\delta_{mj}\frac{\left[(\boldsymbol{\lambda}^{\mathrm{f}})_k\right]^{\frac{1}{2}}}{\left[(\boldsymbol{\lambda}^{\mathrm{f}})_m\right]^{\frac{1}{2}}} + \delta_{kj}\frac{\left[(\boldsymbol{\lambda}^{\mathrm{f}})_m\right]^{\frac{1}{2}}}{\left[(\boldsymbol{\lambda}^{\mathrm{f}})_k\right]^{\frac{1}{2}}}\right]$$

$$= \left[2\left[(\boldsymbol{\lambda}^{\mathrm{f}})_j\right]^{\frac{1}{2}}\left(\boldsymbol{h}_\lambda(\boldsymbol{\lambda}^{\mathrm{f}})\right)_i\right]^{-1}\sum_m(\mathbf{H})_{ij}(\mathbf{H})_{im}\left(\mathbf{P}^{\mathrm{f}}\right)_{jm}\left[(\boldsymbol{\lambda}^{\mathrm{f}})_m\right]^{\frac{1}{2}}. \tag{A1}$$

## Appendix B: Real-world application

We also applied the inflation method to reanalyze the case presented earlier by Bauser et al. (2016), where measurements from 11 TDR probes were assimilated with an EnKF. There, the inflation method confirmed the behavior observed in the small synthetic case presented in this paper. For the details of the real-world case as well as the concept of the closed-eye period please refer to Bauser et al. (2016) or Bauser (2018, Chapter 5), the latter of which also includes the reanalysis of the case.

In this paper, we only show the inflation related to the closed-eye period (Fig. B1), which presents the major challenge to the inflation in that particular application. During this time, preferential flow occurs and the underlying local equilibrium assumption of the Richards equation is violated. With a standard approach, parameters become biased to compensate these errors. To avoid this, Bauser et al. (2016) introduced the closed-eye period, which pauses the parameter estimation and only guides the water content states through times, when assumptions are violated. Compared to the standard approach, this leads to a reduced bias in the parameters, but effectively increases the model errors during the closed-eye period. A strong inflation is required to compensate this error. The inflation method used in Bauser et al. (2016) was just able to accomplish this and the authors were concerned that a too slow adjustment speed of the inflation limits the applicability of the closed-eye period for cases with larger model errors.

Figure B1 confirms the fast adjustment speed of the new inflation method proposed in this paper for the real-world application. The strong required inflation stays well within the closed-eye period. This enables the EnKF to pick up the parameter estimation after the period from a water content state consistent with the TDR measurements and facilitates the use of the closed-eye period.

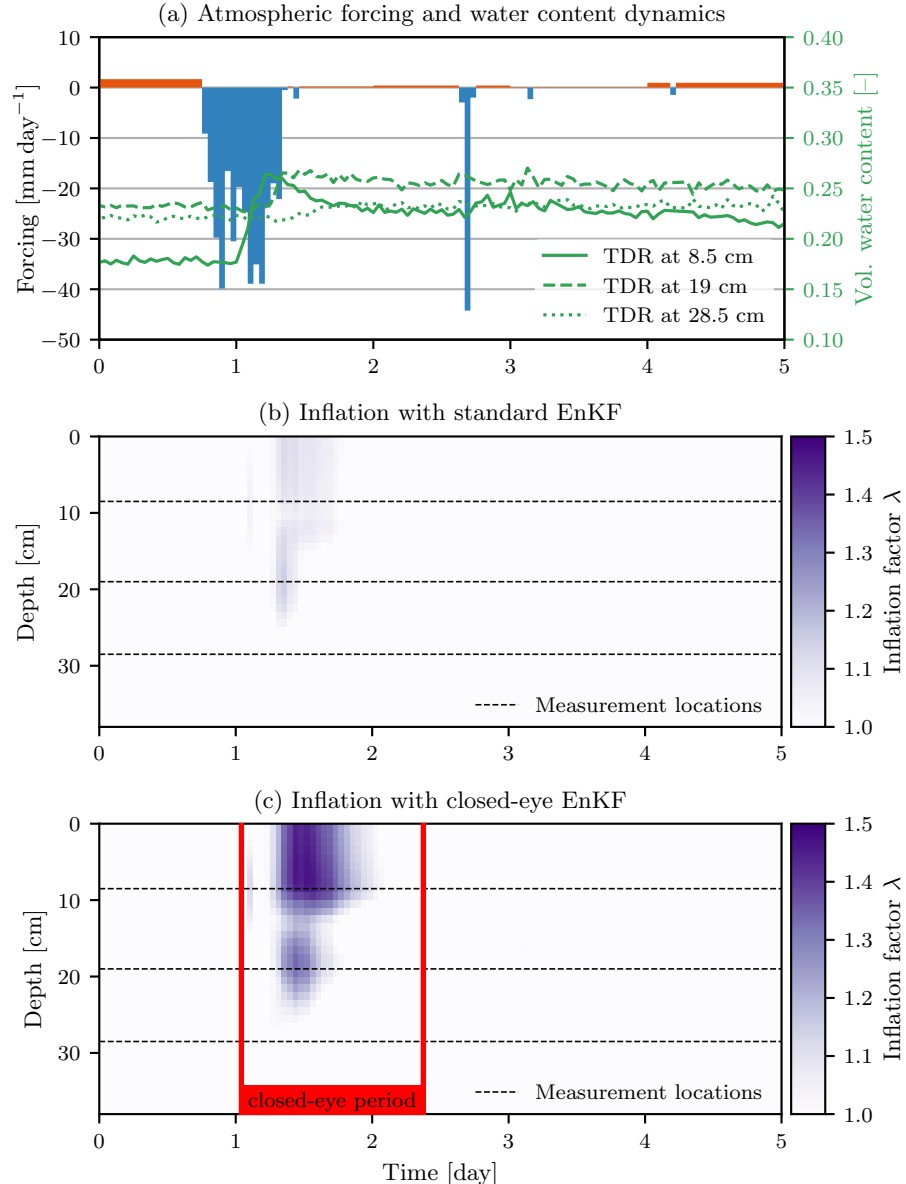

**Figure B1.** Inflation factor for the water content state in a real-world application for a standard and closed-eye EnKF together with atmospheric forcing and corresponding response of measured water contents. The closed-eye period starts when the infiltration front reaches the topmost TDR. During this time the local equilibrium assumption of the Richards equation is violated and a strong inflation is required. The new inflation method allows a fast adjustment of the inflation factors, which enables the EnKF to effectively guide the water content states with the TDR measurements through the closed-eye period. Modified from Bauser (2018).

*Competing interests.* The authors declare that they have no conflict of interest.

*Acknowledgements.* We thank editor Insa Neuweiler and two anonymous reviewers for their comments, which helped to improve this paper.
This research is funded by Deutsche Forschungsgemeinschaft (DFG) through project RO 1080/12-1 and the Ministerium für Wissenschaft,
Forschung und Kunst Baden-Württemberg (Az 33-7533.-30-20/6/2). HGS MathComp provided travel expenses for Hannes H. Bauser and
Daniel Berg.

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
