# Peer review of "Inflation Method for Ensemble Kalman Filter in Soil Hydrology"

_Hydrology and Earth System Sciences, 2018_

## Referee Comment (RC1) · Anonymous Referee #1 · 29 Mar 2018

Summary

The manuscript presents a new approach to handling inflation during Ensemble Kalman filtering, specifically developed for state and parameter estimation in the unsaturated zone. The authors suggests to calculate a dynamic inflation factor that is estimated with a separate Kalman filter within the main EnKF. Apart from theoretical development, the authors also show a test case with good performance.

General comment

I much enjoyed reading the manuscript, which provides an important contribution to the community. The manuscript is well written and well structured, although I think some things should be a bit clearer explained (which is addressed below).

[Figure]

Specific comments Derivations (page 5-6): I find the derivations of the internal Kalman filter sometimes tricky to follow. Primarily I am a bit confused with the second part of eq. 13, which should come from mean inflated forcast vector (Hu-overbar-f-inf ), where the mean of the inflated vector u in turn is the same as the non-inflated mean u-vector (since the inflation, if I get it right, does not alter the mean). Therefore I am not 100% clear on how the lambdas come into the equation (although the equation itself makes sense). Perhaps it would be a good idea to add a more detailed version of the full derivation process in an appendix, where also smaller steps can be shown without creating a too long and tedious paper.

Damping factor: as the authors states in the introduction (page 2, line 31), damping factors are often used instead of inflation, e.g. in lack of better ideas, since it is a rather ad-hoc method. However, despite of now having an advanced and seemingly well-functioning inflation method, the authors still apply a rather strong damping factor to the parameter updates and lambda updates (page 8). Is this still needed?

Inflation factor uncertainty tuning: The authors points out that a higher value is needed for a good performance. The question that arises is whether one can also set it too high (all within reason of course), or if e.g. 1 would, in general, be good enough? Or/similarly, can the authors stress cases where a small inflation factor uncertainty could be clearly beneficial? A short discussion (if possible with some more general recommendation) would clearly increase the transition for other users to use the author's method! (the current discussion discusses more the impact of too low numbers)

In figure 3c one sees an ensemble member with Ksat values around -7, which is clearly far away from the truth and something like an outlier. To me it looks as if the inflation is maintaining this position, as in the corresponding figure with less inflation (figure 4) it is slowly reduced, and with no inflation it is removed (although the full ensemble moves towards it, rather). In a parameter estimation problem, would this really be a wanted behavior? A related question is whether the application of the inflation factor on the parameters or on the states are more influential, hence, if you need really need both

(especially in combination with damping)? Reason for asking is, partly, that in Figure 5 we can see a drifting off of the Ksat already before the rain event which is not visible in Figure 3c, despite the inflation of the parameter in is roughly 1 here. However, a rather substantial inflation is applied to the states (figure 1) during this time, suggesting that there are also effects of state inflation on the parameter estimation.

If I get it right, the inflation is applied before the analysis. Wouldn't this, at least in principle, risk altering the sought correlations between states and parameters, (especially since the inflation is applied to both and the system at hand is nonlinear)? Is this an issue that one should keep in mind or is it, in the opinion of the authors, likely negligible?

Technical stuff

Page 4, line 10: reformulate, reads funny

All figures: the inflated ensemble is almost invisible (I didn't realize until the third read-through that there were also light blue lines), change the color to something better/darker!

---

## Referee Comment (RC2) · Anonymous Referee #2 · 8 Apr 2018

General remarks

This is a well written paper that can be of interest for the hydrological modeling community using data assimilation. It presents the development and application of an adaptive inflation method specifically designed to counteract filter inbreeding in soil hydrology, when increasing the ensemble size is too computationally expensive. The method is applied and proved to work well in a small synthetic test case.

I used to think that inflation methods are not a good way to improve the ensemble Kalman filter, because in principle they are numerical tricks that, in my opinion, depart from the correct theory. When filter inbreeding occurs, the only mathematically consistent way to prevent it would be to increase the ensemble size, thus approaching the limit of an infinite ensemble for which the EnKF is demonstrated to converge to the

Kalman filter. However, I recognize that in some cases this is not practically feasible and in this specific case I like the idea of an adaptive inflation factor that is estimated based on the difference between measurements and mean forecast state, i.e., based on physical considerations and that can give useful insights about the dynamics in the unsaturated zone.

Other than relatively minor edits and comments, reported below, I think there is one main flaw in this paper: the method is demonstrated only in a fairly simple synthetic test case and it does not provide insights on possible issues in more realistic applications. I think an additional test case would make the paper more interesting and its conclusions more robust. The Authors themselves claim that the method has already been tested in a real-world application taken from Bauser et al. (2016): I strongly recommend that such an application be added to this paper and its results discussed in details.

Specific comments

Page 2, line 7: Another relevant reference is Botto et al. (2018) "Multi-source data assimilation for physically-based hydrological modeling of an experimental hillslope", still under discussion in HESS but already fully citable.

Page 3, Eq. 3: Strictly speaking, this is a scaled covariance matrix. The real one should be estimated dividing this by (N-1). Same for the matrix R in Eq. 4.

Page 5, lines 5-6: This statement requires an appropriate reference. Does it refer to a previous study by the same research group?

Page 6, lines 18-19: This is a potentially important issue that is worth of further investigation. Why not repeating the analyses with one or two different values of the seed to actually show this sensitivity?

Page 7, lines 4-8: Please add appropriate reference and explain exactly how the spatial variability is taken into account. Is the parameter $\xi$ a function of space?

Page 10, lines 2-3: This statement is not clear. Can you please clarify, considering also

the additional details required in the previous point?

Page 11, lines 28-30: As suggested above, this should be significantly expanded as a new section of the paper, where to show how the method works in a real-world application.

Page 13, lines 8-12: Why would one want to use a multiplicative parameter in the boundary conditions? Maybe use a different example to illustrate this point.

Page 13, lines 13-20: Similar to the previous one, also this paragraph is not very clear. Please rephrase and give practical examples of errors that can and cannot be represented with the augmented state.

Page 14, line 11: Please see previous point.

---

## Author Comment (AC1) · 26 Apr 2018

***Summary:***
*The manuscript presents a new approach to handling inflation during Ensemble Kalman filtering, specifically developed for state and parameter estimation in the unsaturated zone. The authors suggests to calculate a dynamic inflation factor that is estimated with a separate Kalman filter within the main EnKF. Apart from theoretical development, the authors also show a test case with good performance.*

***General comment:***
*I much enjoyed reading the manuscript, which provides an important contribution to the community. The manuscript is well written and well structured, although I think some things should be a bit clearer explained (which is addressed below).*

**Reply:** We thank the reviewer for the constructive comments and suggestions. We have revised our manuscript taking them into account. The difference pdf is attached and also includes the corrections based on the comments of referee #2. We reference (page, line) to this difference pdf.

**Specific comments**
*Derivations (page 5-6): I find the derivations of the internal Kalman filter sometimes tricky to follow. Primarily I am a bit confused with the second part of eq. 13, which should come from mean inflated forcast vector (Hu-overbar-f-inf), where the mean of the inflated vector u in turn is the same as the non-inflated mean u-vector (since the inflation, if I get it right, does not alter the mean). Therefore I am not 100% clear on how the lambdas come into the equation (although the equation itself makes sense). Perhaps it would be a good idea to add a more detailed version of the full derivation process in an appendix, where also smaller steps can be shown without creating a too long and tedious paper.*

**Reply:** We improved the description for Eq. 13. We incorporated it directly in the text. It now reads (page 5, line 29 – page 6, line 3):

"The measurement error covariance matrix $\mathbf{R}_\lambda$ of $\boldsymbol{d}_\lambda$ can be calculated based on the error covariance matrices of $\boldsymbol{d}$ and $\mathbf{H}\overline{\boldsymbol{u}}_{\text{inf}}^{\text{f}}$,

$$(\mathbf{R}_\lambda)_{ij} = \left| (\mathbf{R})_{ij} + \left( \mathbf{H}\mathbf{P}_{\text{inf}}^{\text{f}}\mathbf{H}^T \right)_{ij} \right|, \tag{13}$$

where the inflated forecast error covariance matrix $\mathbf{P}_{\text{inf}}^{\text{f}}$ can be calculated from the inflation vector and the forecast error covariance matrix by combining Eq. (9) (with vector lambda and entrywise multiplication) and Eq. (3): $\mathbf{P}_{\text{inf}}^{\text{f}} = \mathbf{P}^{\text{f}} \circ [\sqrt{\boldsymbol{\lambda}^{\text{f}}}\sqrt{\boldsymbol{\lambda}^{\text{f}}}^T]$. The entrywise product is denoted by $\circ$ and the entrywise square root of $\boldsymbol{\lambda}$ by $\sqrt{\boldsymbol{\lambda}}$."

We also added an appendix, which shows the derivation of the Jacobian (Eq. 15) in detail (page 14, line 30 – page 15, line 8): "We briefly show the derivation of the Jacobian matrix $\mathbf{H}_\lambda$ for the inflation (Eq. 15). Again, the entrywise product is denoted by $\circ$ and the entrywise square root of $\boldsymbol{\lambda}$ by $\sqrt{\boldsymbol{\lambda}}$:

$$
\begin{aligned}
(\mathbf{H}_\lambda)_{ij} &= \frac{\partial}{\partial\left(\boldsymbol{\lambda}^{\text{f}}\right)_j}\left(\boldsymbol{h}_\lambda(\boldsymbol{\lambda}^{\text{f}})\right)_i \\[2mm]
&= \frac{\partial}{\partial(\boldsymbol{\lambda}^{\text{f}})_j}\left[(\mathbf{R})_{ii}+\left(\mathbf{H}\left[\mathbf{P}^{\text{f}}\circ\left[\sqrt{\boldsymbol{\lambda}^{\text{f}}}\sqrt{\boldsymbol{\lambda}^{\text{f}}}^T\right]\right]\mathbf{H}^T\right)_{ii}\right]^{\frac12}\text{ , with }\mathbf{P}^{\text{f}}\text{ symmetric}\\[2mm]
&= \frac{\partial}{\partial(\boldsymbol{\lambda}^{\text{f}})_j}\left[(\mathbf{R})_{ii}+\sum_m\sum_k(\mathbf{H})_{im}(\mathbf{H})_{ik}(\mathbf{P}^{\text{f}})_{km}\left[(\boldsymbol{\lambda}^{\text{f}})_m\right]^{\frac12}\left[(\boldsymbol{\lambda}^{\text{f}})_k\right]^{\frac12}\right]^{\frac12}\\[2mm]
&= \left[2\left(\boldsymbol{h}_\lambda(\boldsymbol{\lambda}^{\text{f}})\right)_i\right]^{-1}\sum_m\sum_k(\mathbf{H})_{im}(\mathbf{H})_{ik}(\mathbf{P}^{\text{f}})_{km}\frac{\partial}{\partial(\boldsymbol{\lambda}^{\text{f}})_j}\left[\left[(\boldsymbol{\lambda}^{\text{f}})_m\right]^{\frac12}\left[(\boldsymbol{\lambda}^{\text{f}})_k\right]^{\frac12}\right]\\[2mm]
&= \left[2\left(\boldsymbol{h}_\lambda(\boldsymbol{\lambda}^{\text{f}})\right)_i\right]^{-1}\sum_m\sum_k(\mathbf{H})_{im}(\mathbf{H})_{ik}(\mathbf{P}^{\text{f}})_{km}\frac12\left[\delta_{mj}\frac{\left[(\boldsymbol{\lambda}^{\text{f}})_k\right]^{\frac12}}{\left[(\boldsymbol{\lambda}^{\text{f}})_m\right]^{\frac12}}+\delta_{kj}\frac{\left[(\boldsymbol{\lambda}^{\text{f}})_m\right]^{\frac12}}{\left[(\boldsymbol{\lambda}^{\text{f}})_k\right]^{\frac12}}\right]\\[2mm]
&= \left[2\left[(\boldsymbol{\lambda}^{\text{f}})_j\right]^{\frac12}\left(\boldsymbol{h}_\lambda(\boldsymbol{\lambda}^{\text{f}})\right)_i\right]^{-1}\sum_m(\mathbf{H})_{ij}(\mathbf{H})_{im}(\mathbf{P}^{\text{f}})_{jm}\left[(\boldsymbol{\lambda}^{\text{f}})_m\right]^{\frac12}\text{ .}
\end{aligned}
\tag{A1}
$$

"

***Damping factor:*** *as the authors states in the introduction (page 2, line 31), damping factors are often used instead of inflation, e.g. in lack of better ideas, since it is a rather ad-hoc method. However, despite of now having an advanced and seemingly well-functioning inflation method, the authors still apply a rather strong damping factor to the parameter updates and lambda updates (page 8). Is this still needed?*

**Reply:** We agree that the damping factor should not be used to replace an inflation. This can also be seen in the case without inflation, where only a damping factor is used. However, we do consider the damping factor to be useful to alleviate issues of nonlinear relations between measurements and augmented state components, when the dynamics is slow compared to the frequency of measurements. This is typically the case for the parameters: their dynamics is assumed to be constant and their relation to the measurements is often nonlinear. Here, the damping factor can improve the estimation, by reducing the linearized analysis corrections. We see the advantage of the damping factor in this reduced adjustment speed of the parameters. Therefore, we think it is good to show that the inflation method works in combination with a damping factor. We added a comment in the manuscript (page 8, lines 9–10): "The damping factor of 0.3 is applied to the parameters to alleviate issues of nonlinear relations between observations and parameters."

***Inflation factor uncertainty tuning:*** *The authors points out that a higher value is needed for a good performance. The question that arises is whether one can also set it too high (all within reason of course), or if e.g. 1 would, in general, be good enough? Or/similarly, can the authors stress cases where a small inflation factor uncertainty could be clearly beneficial? A short discussion (if possible with some more general recommendation) would clearly increase the transition for other users to use the authors method! (the current discussion discusses more the impact of too low numbers)*

**Reply:** Thank you for pointing this out. Indeed, the value can be chosen too large. The method is limited through the transfer of the linearization in the analysis and additionally through the linearization in the calculation of the Jacobian. Too larger uncertainties of the inflation factor lead to an increasing step size in each analysis step. This can lead to potentially too large inflations. In our case an uncertainty of $\sigma_\lambda = 2$ worked well, while a factor of $\sigma_\lambda = 4$ lead to an overinflation of the indicator parameter $\tau$. However, assigning

an uncertainty of 4 to lambda, which typically ranges between the values 1 and 2, seems a little too large. We extended the discussion accordingly (page 13, lines 16–19):

"Too large uncertainties of the inflation (in our test case $\sigma_\lambda = 4$), where the uncertainty is larger than the typical range for the values of lambda, can also lead to overinflation of weakly correlated parameters. Reasons for this can be the linearizations in the analysis and the calculation of the Jacobian (Eq. 14). This limits the adjustment speed of the inflation."

In this context: We accidentally reported the uncertainty of $\sigma_\lambda^2 = 0.5$ for the test case with slow inflation. However the standard deviation $\sigma_\lambda$ was set to this value. We corrected this throughout the manuscript.

*In figure 3c one sees an ensemble member with Ksat values around -7, which is clearly far away from the truth and something like an outlier. To me it looks as if the inflation is maintaining this position, as in the corresponding figure with less inflation (figure 4) it is slowly reduced, and with no inflation it is removed (although the full ensemble moves towards it, rather). In a parameter estimation problem, would this really be a wanted behavior? A related question is whether the application of the inflation factor on the parameters or on the states are more influential, hence, if you need really need both (especially in combination with damping)? Reason for asking is, partly, that in Figure 5 we can see a drifting off of the Ksat already before the rain event which is not visible in Figure 3c, despite the inflation of the parameter in is roughly 1 here. However, a rather substantial inflation is applied to the states (figure 1) during this time, suggesting that there are also effects of state inflation on the parameter estimation.*

**Reply:** Initially the low value is generated through the random realizations of the Gaussian initial distribution for the parameter. Multiplicative inflations increase the distance of each ensemble member to the mean with the inflation factor. Therefore, this outlier also experiences the strongest inflation, which would need to be adjusted in the analysis. However, in this specific case, this ensemble member corresponds to larger values for the Miller scaling factors at both measurement locations. They effectively increase the hydraulic conductivity and compensate the too small value partially. To remove such outliers faster for example parameters could be intermixed.

We do see the application of the inflation to states and parameters necessary. The state inflation is necessary, to faster adjust the state towards the measurements. This improves the best guess of the states and consequently decreases the influence of the model errors on the parameters. The inflation in the parameters is required to compensate too small uncertainties from model errors, as seen in the example. The differences in the parameter estimation can be explained as well. The saturated hydraulic conductivity drifts away from the true value due to the initially unrepresented heterogeneity. Through the inflation of the Miller scaling factor at the depth of 9.5 cm in the beginning, the Miller scaling factor is adjusted faster towards the true heterogeneity and decreases this effect. In the case without inflation, it does not reach the true value until the rain event and the effect on the saturated hydraulic conductivity remains longer. We changed the description of this case slightly to (page 11, line 7– page 12, line 2): "Again, $K_0$ moves further away from the true value due to the unrepresented heterogeneity and comes closer to the true value during the infiltration event. However, since the Miller scaling factor is not inflated in the beginning, it is adjusted slower. Consequently, the $K_0$ is corrected longer in the wrong direction. The uncertainty eventually becomes too small and in the end the mean is more than 5 standard deviations away from the true value, since the uncertainty cannot be increased any more."

*If I get it right, the inflation is applied before the analysis. Wouldn't this, at least in principle, risk altering the sought correlations between states and parameters, (especially since the inflation is applied to both and the system at hand is nonlinear)? Is this an issue that one should keep in mind or is it, in the opinion of the authors, likely negligible?*

**Reply:** Yes, the inflation is applied before the analysis. However, this does not change the correlations in the forecast ensemble. It only changes the variances. The covariances are adjusted so that the correlations

are kept. Since model errors should be included in the forecast, before the analysis, we apply the inflation there.

**Technical Stuff**

*Page 4, line 10: reformulate, reads funny*
**Reply:** We changed the line to (page 4, lines 10–11): "Based on the measurements, the Kalman gain updates the forecast ensemble to the analysis ensemble:"

*All figures: the inflated ensemble is almost invisible (I didn't realize until the third read-through that there were also light blue lines), change the color to something better/darker!*
**Reply:** We made the ensemble members darker.

[revised manuscript text omitted]

---

## Author Comment (AC2) · 26 Apr 2018

*General remarks*
*This is a well written paper that can be of interest for the hydrological modeling community using data assimilation. It presents the development and application of an adaptive inflation method specifically designed to counteract filter inbreeding in soil hydrology, when increasing the ensemble size is too computationally expensive. The method is applied and proved to work well in a small synthetic test case.*

*I used to think that inflation methods are not a good way to improve the ensemble Kalman filter, because in principle they are numerical tricks that, in my opinion, depart from the correct theory. When filter inbreeding occurs, the only mathematically consistent way to prevent it would be to increase the ensemble size, thus approaching the limit of an infinite ensemble for which the EnKF is demonstrated to converge to the Kalman filter. However, I recognize that in some cases this is not practically feasible and in this specific case I like the idea of an adaptive inflation factor that is estimated based on the difference between measurements and mean forecast state, i.e., based on physical considerations and that can give useful insights about the dynamics in the unsaturated zone.*

**Reply:** We thank the reviewer for the constructive comments and suggestions. We have revised our manuscript taking them into account. The difference pdf is attached and also includes the corrections based on the comments of referee #1. We reference (page, line) to this difference pdf.

We agree that inflation methods depart from the correct theory. To emphasize this we added (page 4, line 24) "heuristic" when introducing the multiplicative inflation.

*Other than relatively minor edits and comments, reported below, I think there is one main flaw in this paper: the method is demonstrated only in a fairly simple synthetic test case and it does not provide insights on possible issues in more realistic applications. I think an additional test case would make the paper more interesting and its conclusions more robust. The Authors themselves claim that the method has already been tested in a real-world application taken from Bauser et al. (2016): I strongly recommend that such an application be added to this paper and its results discussed in details.*

**Reply:** In the real world application described in Bauser et al. (2016), the EnKF is used to estimate the water content state, soil hydraulic parameters, Miller scaling factors and the upper boundary condition for a 1D representation of a soil profile at the Grenzhof test site close to Heidelberg, Germany, based on 11 TDR probes providing water content measurements. To account for uncertainties an assessment of all uncertainties for the specific situation is performed. In a three stage approach (improving the prior, standard EnKF, and closed-eye EnKF) the key uncertainties for this specific situation are reduced, except for an intermittent violation of the local equilibrium assumption in the Richards equation. These times were excluded from the parameter estimation to not incorporate the model error during this time into the parameters. Therefore the forecast during this time has larger errors and in this specific application especially the state inflation during the closed-eye period is important. Figure AC1 shows the inflation of the water content state in with the new inflation method. The main advantage of the new inflation in this case, is that (due to the fast adjustment) the inflation is mainly performed during the closed-eye period only and not continued afterwards.

Including this application into the paper, would increase the length of the paper considerably and require detailed explanations unrelated to the inflation method itself. However, we did not see any conceptually new insights from this application. Therefore, we do not want to include it in the paper, to keep it as concise as possible.

[Figure]

Figure AC1: Inflation factor $\lambda$ in the first layer during the last iteration of the standard EnKF and closed-eye EnKF. The covariance inflation increases the inflation factor around a measurement location when the deviations due to the infiltration reach the corresponding TDR. The deviations and consequently the inflation are stronger for the closed-eye EnKF, as the preferential flow is not incorporated into the parameters.

**Specific comments**

*Page 2, line 7: Another relevant reference is Botto et al. (2018) "Multi-source data assimilation for physically-based hydrological modeling of an experimental hillslop", still under discussion in HESS but already fully citable.*
**Reply:** We completely agree. Botto et al. (2018) must be citet. We became aware of the publication after submission of this manuscript. We included it in the introduction (page 2, line 7 and page 2, line 34).

*Page 3, Eq. 3: Strictly speaking, this is a scaled covariance matrix. The real one should be estimated dividing this by (N-1). Same for the matrix R in Eq. 4.*
**Reply:** Yes, we agree. However, we would like to follow the notation by Evensen (1994) and Burgers et al. (1998), where the overline generally denotes the expectation value. (We added this on page 3, line 28: ". . . , where the overline denotes the expectation value and $\overline{\boldsymbol{u}_k^{\mathrm{f}}}$ is the ensemble mean".) For the measurement

error covariance matrix given in Eq. (4) we actually do not use the matrix calculated from the ensemble of measurement errors, but use the exact diagonal matrix with the measurement uncertainties. Therefore we prefer the more general expression.

*Page 5, lines 5-6: This statement requires an appropriate reference. Does it refer to a previous study by the same research group?*

**Reply:** To our knowledge there exists no study about the inflation method by Anderson (2009) applied to parameters. We are also not aware of an application of the method for parameter estimation in hydrology. Because of that, it is not our goal to perform an in depth analysis of the method. We cannot exclude that it might be possible to use the method, but in our tests, we could see possible over-inflation. Therefore we do not give a general statement in the paper but only state our experience ("However, we experienced possible over-inflation in parameters …"). We would prefer to not work this out in the paper.

Below, we show results obtained with the inflation method by Anderson (2009) to support our statement. Like our proposed inflation, the inflation by Anderson (2009) can be used with a constant uncertainty $\sigma_\lambda$. The values are only roughly related to the values in our method though. Figures AC2, AC3 and AC4 show the results for different choices of $\sigma_\lambda$. They are all analogous to Fig. 3 in the manuscript. We also kept the same limits for the range of the axis for a better comparison.

Figure AC2 shows results for $\sigma_\lambda = 1$. Although the estimates for the mean values are excellent, the inflation is very strong. Especially the Miller scaling factor at 9.5 cm depth is inflated heavily and the good result is not guaranteed. This can be seen in Fig. AC3 ($\sigma_\lambda = 0.5$), where the inflation factor is adjusted slower. It still leads to very strong inflations for the Miller scaling factor at 9.5 cm depth and leads to wrong values in the hydraulic conductivity. Reducing the adjustment speed further (Fig. AC4, $\sigma_\lambda = 0.3$) leads to similar problems as in our inflation method: an overinflation of weakly correlated parameters for slow adjustment speeds.

[Figure]

Figure AC2: Analogous to Figure 3, but with inflation method by Anderson (2009) and $\sigma_\lambda = 1$.

[Figure]

Figure AC3: Analogous to Figure 3, but with inflation method by Anderson (2009) and $\sigma_\lambda = 0.5$.

[Figure]

Figure AC4: Analogous to Figure 3, but with inflation method by Anderson (2009) and $\sigma_\lambda = 0.3$.

*Page 6, lines 18-19: This is a potentially important issue that is worth of further investigation. Why not repeating the analyses with one or two different values of the seed to actually show this sensitivity?*
**Reply:** Different behavior, because of the seed can be indeed an important issue. In this case it effects the behavior of the EnKF itself. The inflation method remains consistent with this behavior. Therefore, we do not want to include further seeds into the manuscript to keep it as concise as possible. We make our statement more clear and change it to (page 6, lines 21–23): "Due to a small ensemble size, the results vary depending on the seed of the random numbers. This however, is related to different performance of the EnKF itself. In simulations (results are not shown), we found that the behavior of the inflation remains consistent."

Figure AC5 shows the parameter estimation (analogous to Fig. 3 in the manuscript) for such a different seed. Especially the initial ensemble influences the correlations and consequently the corrections in the beginning. In this case, the Miller scaling factor at 19.5 cm has stronger correlations and is already corrected towards the true value. This leads to improved parameters at the beginning of the rain event. Therefore the inflation is larger in the beginning and smaller during the rain event.

[Figure]

Figure AC5: Analogous to Figure 3, but with different seed.

*Page 7, lines 4-8: Please add appropriate reference and explain exactly how the spatial variability is taken into account. Is the parameter ξ a function of space?*

**Reply:** Yes, it is. We added the Reference (Miller and Miller, 1956) and extended the explanation (page 7, lines 7–14) to "We additionally consider small scale heterogeneity through Miller scaling. It assumes geometrical similarity. With this the microscopic geometry of the pore space at a macroscopic position is parameterized by a single length scale ξ and the macroscopic heterogeneity field can be generated with a single scalar field of this length scale. Miller and Miller (1956) showed that the hydraulic functions scale with this parameter according to

$$K(\theta) = K^*(\theta) \cdot \xi^2 \,, \tag{22}$$

$$h_{\mathrm{m}}(\theta) = h_{\mathrm{m}}^*(\theta) \cdot \frac{1}{\xi} \,, \tag{23}$$

where the functions $K(\theta)$ and $h_m(\theta)$ are defined at a reference point $^*$ with Miller scaling parameter $\xi = 1$ and from there are projected to all locations."

We additionally changed the explanation how the Miller scaling is included in the augmented state to (page 7, lines 22–24): "We reduce the description of the heterogeneity to these two parameters. The full function of the scaling factors is calculated by linearly interpolating between the measurement positions and constantly extrapolating to the boundaries."

*Page 10, lines 2-3: This statement is not clear. Can you please clarify, considering also the additional details required in the previous point?*

**Reply:** We extended the explanation to (page 8, lines 30–33): "This is due to the chosen interpolation of the Miller scaling factors. Through the interpolation between the measurement locations and extrapolation to the boundaries changes, the dynamics changes at the measurement locations. During the infiltration the dynamics is mainly influenced by the water content above and the correlations to these locations are stronger."

*Page 11, lines 28-30: As suggested above, this should be significantly expanded as a new section of the paper, where to show how the method works in a real-world application.*

**Reply:** Since the application to the real world case did not show any new conceptual insights, we did not include a new section here. Our goal is to keep the paper as concise as possible.

*Page 13, lines 8-12: Why would one want to use a multiplicative parameter in the boundary conditions? Maybe use a different example to illustrate this point.*

**Reply:** We see the boundary condition as a good example, where correlations could drop very rapidly. Different estimations of the boundary condition were included for example in Bauser et al. (2016) or Jaumann and Roth (2017). We agree that a multiplicative factor to the boundary condition is a very specific example. Therefore we generalize the statement to (page 13, lines 20–21): "An example could be a parameter only acting on an infiltration boundary condition."

*Page 13, lines 13-20: Similar to the previous one, also this paragraph is not very clear. Please rephrase and give practical examples of errors that can and cannot be represented with the augmented state.*

**Reply:** We rewrote the paragraph to (page 14, lines 3–15): "The method is in principle capable of compensating unrepresented model errors. However, it relies on correlations calculated from the forecast ensemble of the augmented state. If parameters have correlations to measurement locations with underestimated forecast uncertainties, the inflation will keep increasing the parameter spread until the forecast uncertainties are increased sufficiently. Therefore the correlations have to contain useful information. This means that inflating

the parameters based on their correlations to measurement locations has to increase the forecast spread at these measurement locations. If the parameters have an insufficient influence on the state uncertainty an over-inflation of the parameters can occur. An example are measurements with underestimated measurement uncertainties and short time between measurements compared to the timescale of the dynamics. Then the parameters are not able to increase the state uncertainty in the short forecast time between measurements and the forecast dynamics is not able to represent the measurement noise. If such errors occur intermittently, e.g., the closed-eye period as proposed by Bauser et al. (2016) could be used to bridge these times. A rather heuristic solution could be a decay of the inflation factor towards values of 1, as already proposed by Anderson (2009)."

*Page 14, line 11: Please see previous point.*
**Reply:** We changed the sentence (analogous to the explanation above) to (page 14, lines 26–27): "
[revised manuscript text omitted]

---

## Author Response (AR2)

**Answer to Review by Referee #2**

*The paper has been improved compared to the previous version. However, it seems the Authors are not willing to include an additional test case based on real data taken from Bauser et al. (2016), on ground that it would not give new conceptual insights and it would make the paper too long. I disagree. In my opinion, it is very important to show that the proposed approach can work even in real-world applications. The literature is full of approaches shown to work very well for synthetic test cases but never really applied with real-world data. Given that the simulations are done already, the amount of work required to include these new results in the paper would be relatively limited. Also, new insights not necessarily related directly to the inflation, but linked to the physics of the problem would be more than welcome. These additions would not make the paper too long, but rather more interesting and robust, resulting in a valuable contribution to the data assimilation literature in soil hydrology.*

**Reply:** We agree with the reviewer that it is important that methods do work with real-world data. As stated in the manuscript (with the main result shown in the previous answer), the method **does** work on the real-world case by Bauser et al. (2016). It is thus not so much a lack of willingness to do the demonstration, which we already did, but to prove our words to be true in this paper. We include the key phase of Bauser et al. (2016) in the modified manuscript.

In the PhD thesis by Bauser (2018, Chapter 5), the real-world case is repeated with the proposed inflation method. Mostly, the case is less demanding for the inflation method than the presented synthetic case in this paper. This is due to a detailed assessment and representation of relevant uncertainties as well as a three stage approach (improving the prior, standard EnKF and closed-eye EnKF). Particularly, improving the prior of Miller scaling factors reduced the requirements for the inflation method. However, Bauser et al. (2016) introduced the concept of the closed-eye period to bridge times when the model errors cannot be represented. During this time the inflation method is relevant, since the applicability of the closed-eye period can be limited by the adjustment speed of the inflation factor. The new inflation method is capable of increasing (and decreasing) the inflation within the closed-eye period rapidly and makes the closed-eye period more applicable. Note that during the closed-eye period the parameters are kept constant and therefore the transfer of the inflation to the parameters is not of importance. We include this finding in the manuscript. Apart from this the insights found in Bauser et al. (2016) could be confirmed. We would find it impertinent to basically repeat the findings of a previously published paper again. We refer to Bauser et al. (2016) for the detailed explanations describing the real-world case the closed-eye period. For the readers interested in the full reanalysis, we refer to Bauser (2018).

We changed the manuscript to (page 12, line 1 - page 13, line 7):

[revised manuscript text omitted]